# The Role of Fermented Dairy Products on Gut Microbiota Composition

**Adam Okoniewski [1], Małgorzata Dobrzyńska [1], Paulina Kusyk [1], Krzysztof Dziedzic [2], Juliusz Przysławski [1] and Sławomira Drzymała-Czyż [1,*]**

[1] Department of Bromatology, Poznan University of Medical Science, Rokietnicka 3 Street, 60-806 Poznan, Poland

[2] Department of Food Technology of Plant Origin, Poznań University of Life Sciences, Wojska Polskiego 28 Street, 60-637 Poznań, Poland

[*] Correspondence: drzymala@ump.edu.pl; Tel.: +48-61-641-83-83

**Abstract:** Milk and dairy products are among the most important foods in the human diet. They are natural and culturally accepted and supply the human body with microorganisms that modulate the intestinal microflora. Improper lifestyles, highly processed diets, and certain drugs may contribute to adverse changes in the composition of the gut microflora. These changes may lead to dysbiosis, which is associated with the pathogenesis of many gastrointestinal diseases. This review aims to determine the effect of fermented milk products on the composition of the gut microbiota and their possible support in the treatment of dysbiosis and gastrointestinal diseases. While most research concerns isolated strains of bacteria and their effects on the human body, our research focuses on whole fermented products that contain complex mixtures of bacterial strains.

**Keywords:** microbiota; microbiome; fermented milk product; dairy product; dysbiosis; eubiosis

## 1. Introduction

In recent decades, the human diet and lifestyle have changed, with the Western diet increasingly dominated by highly processed products that are rich in preservatives, fats and simple sugar and low in fiber and other nutrients [1,2]. This may modulate the composition of the gut microbiota. Further, available drugs (antibiotics, proton pump inhibitors) may also affect the composition of the intestinal microbiota. These changes may contribute to dysbiosis, which is currently associated with the pathogenesis of many diseases [3–5].

One of the non-pharmacological ways to enrich the depleted gut microbiota is to consume naturally fermented dairy products, which are rich in beneficial microorganisms [6,7].

Fermented dairy products are naturally rich in postbiotics, which are defined as preparations of non-living microorganisms and/or their components that confer a health benefit on the host. They may be antioxidant, anti-inflammatory, anti-bacterial, anti-cancer, and immunomodulatory, and they may support the treatment of obesity, dyslipidemia, or hypertension. These multidirectional and pleiotropic effects result from the multitude of compounds that are classified as postbiotics: enzymes, lipids, proteins, saccharides, vitamins, coenzymes, organic acids, complex particles, and others. Among the postbiotics that may be present in fermented milk products are, e.g., aminobutyric acid (GABA), amino acids such as ornithine and tryptophan, and lactic acid. On the other hand, postbiotics may be used in the production technology of dairy products as natural preservatives, which are often resistant to high temperatures. An example of such a bacteriocin is nisin-a lantibiotic, which is produced by selected strains of *Lactococcus lactis*. Nisin inhibits the growth of mainly Gram-positive bacteria and is approved for use in food preservation. It seems that postbiotics may become a new source of functional foods [8].

This review aimed to determine the effect of fermented milk products as a typical functional food on human gut microbiota composition. Specifically, we aimed to determine

the possible impact of nutritional treatment with fermented milk products on the gut microorganisms, including dysbiosis, and to consider the treatment of gastrointestinal diseases. Currently, available review studies focused mainly on isolated strains of bacteria and their beneficial effects on the human body. Our research focuses on the whole fermented products, not on individual isolated strains. The databases PubMed, Scopus, Cochrane Library, Web of Sciences, and Embase were searched up to 18 January 2023 using the following phrases: "fermented milk product" and "gut microbiota" or "fermented dairy product" and "gut microbiota" or "fermented milk product" and "intestinal microbiota" or "fermented dairy product" and "intestinal microbiota" to identify relevant English-language articles. The review used only original articles assessing the effects of fermented milk products on the gut microbiota. We did not evaluate studies that were concerned with isolated bacterial strains. Additionally, the reference lists of retrieved articles were screened manually to find potential relevant literature. The search strategy is presented in Figure 1.

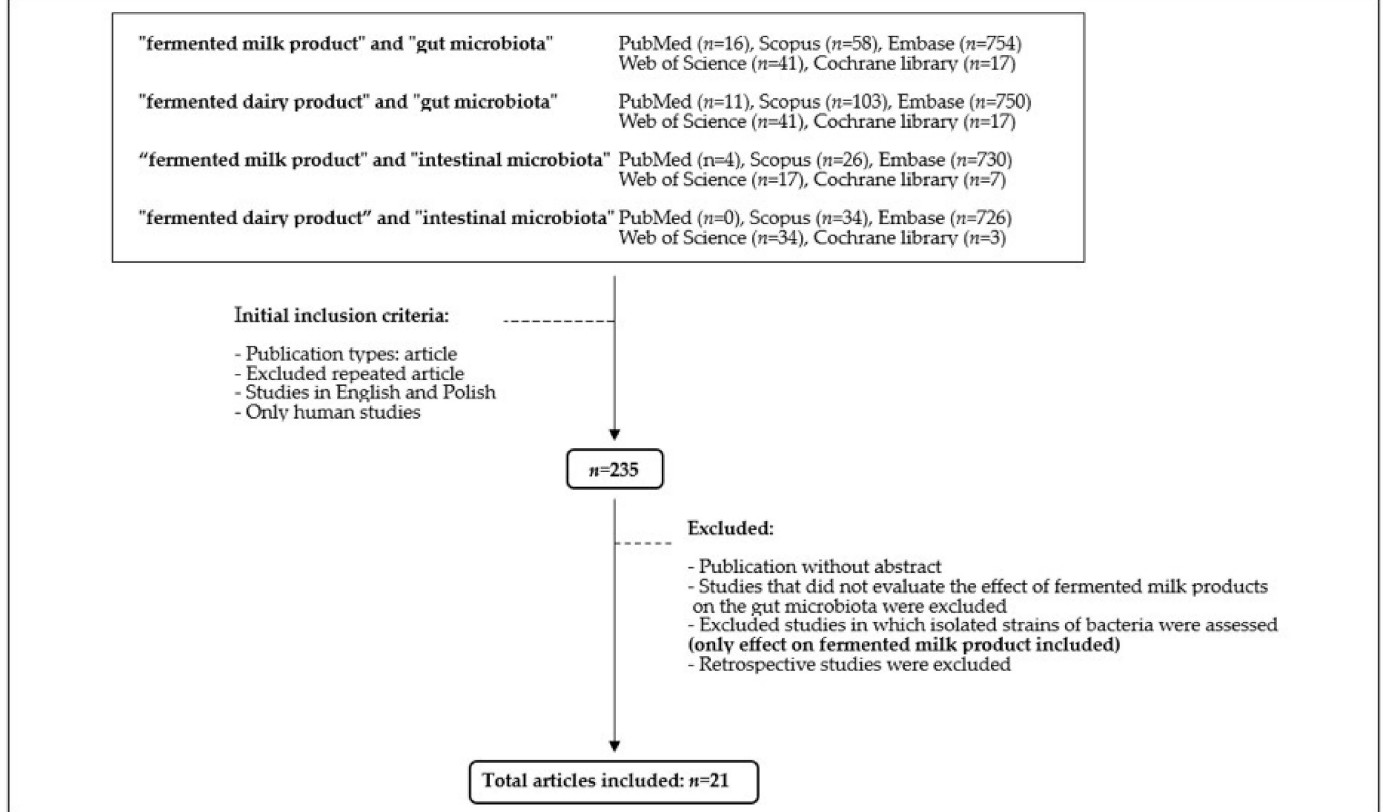

**Figure 1.** Search strategy.

## 2. Fermented Milk Products

Milk and fermented milk products have a long history of use dating back to the seventh millennium BC [9–11] and have been eaten since the domestication of ruminants. For at least 10,000 years, they have been an important component of the daily diets of people around the world, especially in the case of pastoral populations, where dairy products have been and are the foundation of daily diets [12–14]. In recent decades, technological innovation has led to a wide range of dairy products, some with ingredients such as fat and lactose removed or reduced and others fortified with ingredients such as iron, sterols, and vitamin D. Increased awareness of the link between diet and health has increased the demand for certain types of products, such as those low in fat and calories and products to which vitamins and minerals have been added [15].

Fermented milk products include dairy foods that have been fermented by suitable microorganisms which convert some of the lactose to lactic acid, resulting in reduced pH

with or without coagulation [15]. Fermented milk is prepared from whole milk, mostly skimmed or fully skimmed and concentrated, or from a milk substitute of partially or fully skimmed milk powder, partially or fully skimmed, that is pasteurized or sterilized and subjected to fermentation [16,17]. These products are usually made from cow, goat, sheep, or buffalo milk, as well as milk from other animals, such as camels, mares, and donkeys [15].

### 2.1. Composition of Various Kinds of Milk Used to Produce Fermented Milk Products

Milk and fermented milk products are a major source of dietary energy, protein, and fat, contributing an average of 134 kcal of energy/per person/day [15,18]. However, this varies by geographic region, with the percentage of dietary energy from milk in daily food rations in Asia and Africa lower than that in Europe and Oceania, supplying 3% and 8–9% of energy in the diet, respectively.

Milk is composed of approximately 75–91% water, 0–9% fat, 1–6% protein, 3–7% lactose, and minerals and vitamins, with some variation depending on the animal from which it is obtained [19–21]. The main minerals found in milk are calcium and phosphorus, with smaller amounts of potassium, magnesium, zinc, and selenium [22,23]. Milk is a valuable source of water-soluble B vitamins (especially riboflavin and B12) and fat-soluble vitamins (such as vitamins A, D, and E), which are directly related to the lipid content. It also contains immunoglobulins, hormones, growth factors, cytokines, nucleotides, peptides, polyamines, enzymes, and other bioactive peptides [24]. The composition of the most common kinds of milk used to produce fermented milk products is presented in Table 1.

**Table 1.** The composition of milk from various animals (per 100 g of fresh milk) [15,25].

| Nutrients | Cow Milk | Goat Milk | Sheep Milk | Buffalo Milk | Mare Milk | Camel (Dromedary) Milk | Donkey Milk | Yak Milk |
|---|---|---|---|---|---|---|---|---|
| Energy (kcal) | 59–66 | 57–69 | 93–108 | 71–118 | 42–50 | 44–79 | 32–51 | 87–91 |
| Energy (kJ) | 247–274 | 243–289 | 388–451 | 296–495 | 177–210 | 185–332 | 135–215 | 349–382 |
| Water (g) | 87.3–88.1 | 86.4–89.0 | 80.7–83.0 | 82.3–84.0 | 87.9–91.3 | 88.7–89.4 | 89.2–91.5 | 75.3–84.4 |
| Protein (g) | 3.2–3.4 | 2.9–3.8 | 5.4–6.0 | 2.7–4.6 | 1.4–3.2 | 2.4–4.2 | 1.4–1.8 | 4.2–5.9 |
| Fat (g) | 3.1–3.3 | 3.3–4.5 | 5.8–7.0 | 5.3–9.0 | 0.5–4.2 | 2.0–6.0 | 0.3–1.8 | 5.6–9.5 |
| Lactose (g) | 4.5–5.1 | 4.2–4.5 | 4.5–5.4 | 3.2–4.9 | 5.6–7.2 | 3.5–4.9 | 5.9–6.9 | 3.3–6.2 |
| Calcium (mg) | 91–120 | 100–134 | 170–207 | 147–220 | 76–124 | 105–120 | 68–115 | 119–134 |
| Iron (mg) | Tr.–0.2 | Tr.–0.6 | Tr.–0.1 | 0.2 * | Tr.–0.2 | 0.2–0.3 | | 0.2–1.0 |
| Magnesium (mg) | 10–11 | 13–14 | 18 * | 2–16 | 4–12 | 12–14 | 4 * | 8–12 |
| Phosphorus (mg) | 84–95 | 90–111 | 123–158 | 102–293 | 43–83 | 83–90 | 49–73 | 77–135 |
| Potassium (mg) | 132–155 | 170–228 | 120–187 | 112 * | 25–87 | 124–173 | 50 * | 83–107 |
| Sodium (mg) | 38–45 | 32–50 | 30–44 | 47 * | 13–20 | 59–73 | 22 * | 21–38 |
| Zinc (mg) | 0.3–0.4 | 0.1–0.5 | 0.5–0.7 | 0.5 * | 0.2–0.3 | 0.4–0.6 | | 0.7–1.1 |
| Copper (mg) | Tr. | Tr.–0.1 | 0.1–0.1 | | Tr.–0.1 | 0.1–0.2 | | 0.4 * |
| Selenium (µg) | 1.0–3.7 | 0.7–1.4 | 1.7 * | | | | | |
| Manganese (µg) | 4–10 | Tr.–18 | Tr.–18 | | | 60–180 | | |
| Vitamin A (µg) | 30–46 | 30–74 | 64 * | 69 * | | | | 14 * |
| Vitamin E (mg) | 0.1–0.1 | Tr.–0.1 | 0.1–0.1 | 0.2–2.0 | | | | Tr |
| Thiamin (mg) | Tr. | Tr.–0.1 | 0.1–0.1 | 0.1 * | Tr. | | 0.1 * | 0.1 * |
| Riboflavin (mg) | 0.2–0.2 | Tr.–0.2 | 0.3–0.4 | 0.1 * | Tr. | 0.1 * | Tr. | 0.1 * |
| Niacin (mg) | 0.1–0.2 | 0.1–0.3 | 0.4–0.4 | 0.2 * | 0.1 * | | 0.1 * | Tr. |

**Table 1.** *Cont.*

| Nutrients | Cow Milk | Goat Milk | Sheep Milk | Buffalo Milk | Mare Milk | Camel (Dromedary) Milk | Donkey Milk | Yak Milk |
|---|---|---|---|---|---|---|---|---|
| **Pantothenic acid (mg)** | 0.3–0.6 | 0.3–0.4 | 0.4–0.5 | 0.2 * | | | | |
| **Vitamin B6 (mg)** | Tr.–0.1 | 0.1–0.1 | 0.1–0.1 | 0.3 * | | | | Tr. |
| **Folate (µg)** | 5.0–8.0 | Tr.–1.0 | 5.0–7.0 | 0.6 * | | | | |
| **Biotin (µg)** | 1.4–2.5 | 2.0–3.0 | 2.5–2.5 | 13.0 * | | | | |
| **Vitamin B12 (µg)** | 0.3–0.9 | Tr.–0.1 | 0.6–0.7 | 0.4 * | | | | |
| **Vitamin C (mg)** | Tr.–2.0 | 1.1–1.3 | 4.2–5.0 | 2.5 * | 1.7–8.1 | 2.5–18.4 | | |
| **Vitamin D (µg)** | 0.1–0.3 | 0.1–0.1 | 0.2–0.2 | | | | | 0.2 * |

Tr.—trace amount (<0.05); *—average value; blank spaces indicate that no data were available.

It is worth noting that both milk protein and lactose in fermented milk products are more easily absorbed than in plain milk, as the bacterial proteolytic system partially degrades proteins. The lactose content is lower because some is converted into lactic acid and/or alcohol. Moreover, yogurt and fermented milk may contain more folic acid than plain milk because some strains of lactic acid bacteria also synthesize folate. Fermentation not only makes milk more digestible but also increases the shelf life and microbiological safety of products [15].

### 2.2. Types of Fermented Milk Products

According to the European Food Information Council (EUFIC), there are over 3500 traditional, fermented food products in the world. The principal types are yogurt, kefir, soured milk, and kumis, with lesser and regional amounts of other products (e.g., Långfil, Villi) [26]. These are classified into three different types, depending on the type of fermentation:

1. Products of lactic fermentation, where strains of mesophilic or thermophilic lactic acid bacteria are used (e.g., yogurt).
2. Products obtained through alcohol-lactic fermentation involving yeast and lactic acid bacteria (e. g., kefir, kumis).
3. Products with mold growth in addition to the fermentation types above (e.g., viili) [27].

Each type of fermented milk product has a specific characteristic composition, taste, and texture, depending on the type of milk, the starter cultures, and the method of preparation (Table 2). There are four categories of fermented products based on these characteristics: moderately sour types with a pleasant aroma associated with diacetyl, sour and very sour types owing to high acid production, ethanol in addition to lactic acid, and probiotic fermented milk products [17,28].

**Table 2.** Characteristics of selected fermented milk products [17,27,29–37].

| Fermented Milk Product | Type of Milk | Fermentation Culture | Basic Product Characteristic |
|---|---|---|---|
| Yogurt | All types, especially cow, goat, sheep, and buffalo milk | *Streptococcus* (*Sc.*) *thermophilus* and *Lactobacillus* (*Lb.*) *delbrueckii* sp. *Bulgaricus* | Tart flavor and texture related to the fermentation of sugars in milk and the production of lactic acid. |
| Kefir | Especially from cows, goats, or sheep milk | *Lc. lactis* subsp. *lactis*, *Lc. lactis* subsp. *cremoris*, citrate-positive *Lc. lactis*, *Ln. mesenteroides* subsp. *cremoris*, *Ln. mesenteroides* subsp. *dextranicum*, *Sc. thermophilus*, *Lb. delbrueckii* subsp. *bulgaricus*, *Lb. acidophilus*, *Lb. helveticus*, *Lb. kefir*, *Lb. kefiranofaciens*, *Kluyveromyces marxianus*, *Saccharomyces* spp. | From the North Caucasian regions and Turkey; contains the characteristic microflora of kefir grains; sour, bitter, and slightly carbonated taste similar to drinkable yogurt. The starter culture used affects the viscosity and chemical composition of kefir. |

**Table 2.** *Cont.*

| Fermented Milk Product | Type of Milk | Fermentation Culture | Basic Product Characteristic |
|---|---|---|---|
| Kumis | Mare and donkey milk (Columbian kumis from cow milk) | *Lb. acidophilus, Lb. delbrueckii* subsp. *bulgaricus, Saccharomyces lactis, Kluyveromyces marxianus Pichia membranaefaciens Saccharomyces cerevisiae* | Traditionally produced by fermenting raw milk with yeast and lactic acid bacteria. |
| Långfil | Cow milk | *Lc. lactis* ssp. | Swedish ropy sour milk requires a low acidification temperature and long maturation; mildly acidic with a chewy and cohesive texture. |
| Viili | Cow and other milk | *Lc. lactis, Geotrichum candidum* | Finnish ropy milk product; semi-solid structure with a sharp taste and good diacetyl flavor. |
| Grassis | Camel milk | *Lc. paracasei* subsp., *Lc. plantarum, Lc. lactis, Enterococcus* spp., and *Leuconostoc* spp. | Consumed in various regions of the Sudan; obtained by semi-continuous or fed-batch fermentation process in large skin bags containing a large quantity of previously soured product. |
| Filmjölk | Cow milk | *Lc. lactis*, and *Ln. mesenteroides* subsp. *Cremoris* | Traditional fermented milk products from Sweden; a mild and slightly sour taste. |
| Buttermilk | All types of milk, especially cow milk | *Lc. lactis* subsp. *lactis, Lc. lactis* subsp. *cremoris, Lc. lactis*, and *Ln. mesenteroides* subsp. *Cremoris* | Obtained during the production of butter, containing water-soluble milk components and bioactive material derived from milk fat membrane globules. |
| Dadih | Buffalo milk | *Lb. casei* subsp. *casei, Ln. paramesenteroides, Lb. plantarum, Lc. lactis* subsp. *lactis, Lc. lactis* subsp. *cremoris, citrate-positive Lc. lactis, Enterococcus faecium* | Traditional fermented milk popular in West Sumatra (Indonesia); -produced by pouring fresh, raw, unheated milk into a capped bamboo tube and allowing it to ferment spontaneously at room temperature for a few days. |
| Dahi (Curud) | Cow milk, and sometimes buffalo, yak, or goat milk | *Sc. thermophilus Lb. delbrueckii* subsp. *bulgaricus* or *Lc. lactis* subsp. *lactis, Lc. lactis* subsp. *cremoris, citrate-positive Lc. Lactis* | Popular throughout the Indian subcontinent (around 90% of the total fermented milk products produced in India); obtained from pasteurized or boiled milk fermented with a culture. |
| Yakult | Cow milk | *Lb. casei* subsp. *casei* | Japanese sweetened fermented milk; consists of water, skimmed milk, glucose-fructose syrup, sucrose and bacterial strains. |
| Kurut | Yak milk and other animal milk | *Lb. delbrueckii* and *Lb. helveticus* | Traditional product in northwestern China; obtained by drying yogurt or ayran after filtration with the addition of salt. |
| Tarag | Goat and cow milk | *Lb. helveticus* and *Lb. delbrueckii* ssp. *bulgaricus* | Traditional product in Mongolia and China; produced from raw whole milk by backslopping method. |
| Leben | Cow, goat, sheep, and camel milk | *Lc. lactis* and *Sc. thermophilus, Enterococcus faecium* | Traditional fermented milk from the Middle East and North Africa; produced from raw milk |
| Khoormog | Camel milk | *Lc. helveticus, Lc. kefiranofaciens* and *Lc. delbrueckii* | Mongolian traditional food; a sour and alcoholic taste from raw milk. |

In addition, fermented milk products can be divided into concentrated, flavored, and fermented milk drinks. The concentrated drink is a fermented milk product in which the protein has been increased before or after fermentation to a minimum of 5.6%—for example, Leben. Flavored milk products are composite milk products containing a maximum of 50% of non-dairy ingredients (such as nutritive and non-nutritive sweeteners, fruits, juices,

pulps, cereals, chocolate, nuts, coffee, spices, and other harmless natural flavorings). The non-dairy ingredients can be added before or after fermentation. Fermented milk drinks contain a minimum of 40% fermented milk, as well as other microorganisms, in addition to the specific starter cultures [22]. Yogurt, eaten together with other ingredients such as fruit, may provide combined health benefits through potential prebiotic and probiotic effects. Yogurts are a potential source of probiotics, while fruits are rich in fiber, antioxidants, vitamin C, and potential prebiotics. Yogurt and fruit, separately, may have a protective effect against specific diet-related diseases, such as obesity and type 2 diabetes [38].

## 3. Gut Microbiota

The human body is colonized by a huge number of microorganisms, both internally and externally [39,40]. Microbes colonize the human body immediately after birth and persist until death [41,42]. They account for over 1 kg of human body weight and at least ten times the number of human eukaryotic cells. Most microorganisms colonize the digestive tract, constituting the so-called gut microbiota, and include bacteria, fungi, eukaryotes, viruses, phages, and archaea. Humans and microbes have developed complex symbiotic relationships of co-evolution, co-adaptation, and interdependence. The proper term for the relationship between humans and their microbiota is mutualism (both humans and microbes have their benefits) [40]. As already mentioned, the digestive tract is the most inhabited system, but the degree of colonization is not uniform. Differences in the environments in the parts of the digestive tract cause the diversity of the composition of microorganisms [39,43–46], as set out below:

Oral cavity—the number of microorganisms reaches $10^8$ CFU/mL, predominately *Streptococcus*, *Peptococcus*, *Staphylococcus*, *Bifidobacterium*, *Lactobacillus*, and *Fusobacterium* genera.

Stomach and duodenum—the secretory effect of the stomach (lowering the pH to 1–2) and duodenum results in the death of most bacteria, so the number of bacteria in the stomach is less than $10^1$–$10^3$ CFU/mL (mainly *Lactobacillus*, but also *Helicobacter pylori*), and the number of bacteria in the duodenum is $10^1$–$10^4$ CFU/mL (*Lactobacillus* and *Streptococcus* predominate).

Jejunum and ileum—the number of microorganisms increases to $10^5$–$10^7$ CFU/mL, and these are mainly bacteria of the genera *Bacteroides*, *Lactobacillus*, and *Streptococcus*. In the ileum, the number of microorganisms reaches $10^7$–$10^8$ CFU/mL, with the predominant genera of *Bacteroides*, *Clostridium*, *Enterococcus*, *Lactobacillus*, and *Veillonella*, and species from the *Enterobacteriaceae* family.

Large intestine—the most metabolically active organ of the human body with approximately 70% of all microorganisms colonizing the digestive tract inhabiting the colon. The colonization density of the large intestine is $10^{10}$–$10^{12}$ CFU/mL, predominately *Bacillus*, *Bacteroides*, *Clostridium*, *Bifidobacterium*, *Enterococcus*, *Eubacterium*, *Fusobacterium*, *Peptostreptococcus*, *Ruminococcus*, and *Streptococcus* [47–49].

### 3.1. Factors Affecting Variability in Gut Microbiota Composition

The gut microbiota consists of five basic types of bacteria (*Firmicutes*, *Bacteroidetes*, *Actinobacteria*, *Proteobacteria*, and *Fusobacteria*), constituting up to 90% of the gut microbiome. However, this proportion may not be similar in all individuals, due to interpersonal differences and the multitude of factors explained in this section [50,51].

It is impossible to precisely determine what microorganisms and in what numbers should be present in the human intestines, or what "profile" of microbes is indicated or not recommended for health. The composition of the intestinal microflora is individualized [52]. Nevertheless, the presence of certain species may predispose to the development of certain disease entities, e.g., inflammatory bowel disease, cancer, allergies, or obesity [53–57].

Diet has an important influence on gut microbiota composition. To confirm this hypothesis, some researchers sequenced the oral microbiota from the skeletal teeth of people living in different eras, showing that the most significant changes in the human gut microflora took place during two social and nutritional breakthroughs in the history

of mankind: the transition from the Palaeolithic era of hunter-gatherers to the Neolithic era of agriculture (10,000 years ago), with diets rich in carbohydrates, and the beginning of industrialization (about two centuries ago), characterized by diets rich in processed flour and sugar. Further studies have shown differences in the composition of the gut microbiota among different populations, possibly due to the variability in diet and genetics [58–60]. Considering the gut microbiota at the taxonomic level of species, there is considerable variation between individuals, such that the composition of the microbiota can be compared to a fingerprint. Both endogenous and exogenous factors will affect the composition of the microflora. The diversity among individuals is easy to understand when we consider the countless factors that influence the composition of the gut microbial ecosystem. The host's genetic background plays an important role in the first colonizing bacteria through the bacterial attachment sites (pioneer flora) [61–63]. Pioneer flora modulates host gene expression, affecting subsequent microbial flora. In addition, environmental factors, such as age, diet, stress, and medications, strongly affect the human microbiota composition [64–66].

Other factors influencing the microbiome are the maternal vaginal and gut microflora, which may affect the fetal microflora; the composition and development of the child's gut microflora are greatly influenced by the type of childbirth (natural or by caesarean section) and feeding (mother's milk vs. infant formulae). In addition, the composition of the microbiota is affected by therapeutic procedures, hygiene, exposure to the natural environment, and genetic origins, as evidenced by studies on identical twins [61,67–69].

Although long ignored, viruses play an important role in the gut ecosystem, with 90% of the intestinal virome consisting of bacteriophages and the remaining 10% being plant and zoonotic viruses that are constantly introduced with food [70,71].

Factors affecting the development of the human gut microflora are strongly related to child development and adult life. At the taxonomic level, a healthy adult human microbiota consists mainly of *Firmicutes* and *Bacteroidetes*, which may account for 70% of the total microbiota. *Proteobacteria*, *Verrucomicrobia*, *Actinobacteria*, *Fusobacteria*, and *Cyanobacteria* can also be found in lower percentages. Obligate anaerobes dominate and outnumber facultative anaerobes by two orders and aerobes by three orders [72–74].

The gut microbiota has been divided into three main enterotypes, with each enterotype characterized by a relative abundance of one of the following types of bacteria [40,42,75,76]:

- *Bacteroides* (more represented in enterotype 1),
- *Prevotella* (more numerous in enterotype 2),
- *Ruminococcus* (dominant in enterotype 3).

The prevalence of a particular enterotype may depend on long-term dietary habits; indeed, a high-fat and high-protein diet promotes the growth of enterotypes 1 and 3, while a high-carbohydrate diet supports the growth of enterotype 2. Recent findings suggest that the composition of the gut microbiota is also influenced by short-term dietary changes [40]. Nevertheless, these three variants seem to be independent of body mass index, age, gender, or nationality [42].

Once the composition of the intestinal microflora is established, it theoretically remains stable throughout adult life. Some differences can be observed between the intestinal microbiota of the elderly and young adults, mainly due to the dominance of the genera *Bacteroides* and *Clostridium* in the elderly and *Firmicutes* in young adults [42]. However, it should be emphasized that diet, stress, and medications (antibiotics, proton-pump inhibitors, opioids) can significantly affect the microbiota profile [77–79].

### 3.2. Main Functions of the Gut Microbiota

The gut microbiota plays an important role in maintaining health, mainly participating in the development of immunity and regulating several basic metabolic pathways [5,47,52]. The state of eubiosis and dysbiosis of the intestinal microflora also strongly affects health and disease [41,80,81].

Quantitative and/or qualitative changes in the gut microflora impair this homeostasis, leading to the development of diseases related to the gut microflora, such as functional

diseases of the gastrointestinal tract, infectious diseases of the intestine, inflammatory bowel diseases, liver diseases, gastrointestinal cancers, obesity and metabolic syndrome, allergies, diabetes, and autism [47,53,54,80,82]. The distal section of the human intestine is considered an anaerobic bioreactor with metabolic activity comparable to that of the liver; therefore, the microbiota can be regarded as an organ with specific functions [83,84]. An organ that consumes, conserves, and redistributes energy undergoes physiologically important chemical changes and can maintain and repair itself by self-replication.

The mechanisms of action of probiotic strains [8,85–89] include the following:

1.  Protective function; One of the leading defence mechanisms is the occupation of an ecological niche, which makes it difficult for pathogenic bacteria to reach the intestinal epithelial layer. At the same time, numerous commensal bacteria block receptors are recognized by pathogenic bacteria. An example is *Lb. plantarum*, which uses mannose receptors for adhesion. The same receptors are necessary for the adhesion of enteropathogenic *Escherichia coli* strains. Moreover, commensal bacteria compete with pathogens for nutrients and production of compounds with bacteriostatic/bactericidal activity (bacteriocins, organic acids, hydrogen acid, compounds of the lactoperoxidase system, and others), modification of the intestinal environment to make it unfavorable for the development of harmful microorganisms (lowering pH), thereby maintaining the continuity of the gastrointestinal mucosa: stimulating the secretion of mucin "sealing" the intestinal epithelium and production of short-chain fatty acids and polyamines (regeneration of the epithelium and the effect on cell maturation).

2.  Digestive function: Gut microbiota is involved in the digestion of numerous compounds that are otherwise inaccessible to humans, such as cellulose, pectin, or lignin. These compounds are converted into simple sugars or short-chain fatty acids. An interesting example here may be *Bifidobacterium longum* subs. *infantis* colonizing the intestines of newborns and breaking down HMO (human milk oligosaccharider) sugars not broken down by human digestive enzymes. However, bacteria provide not only nutrients but also vitamins necessary for humans, such as K, B1, B6, B12, or folic acid.

3.  Immune function and Stimulation of the immune system: Probiotic bacteria do not differ significantly from pathogenic bacteria, and ingredients such as lipopolysaccharide (LPS), peptidoglycan, or lipoteichoic acids are recognized by the TLR (tool-like receptor) in the same way. These receptors are involved in stimulating the immune response by promoting the production of pro-inflammatory cytokines (such as TNF-$\alpha$ or IL-1, 6, 8, 12). The NF-$\kappa$B transcription factor is also activated, leading to, e.g., production of anti-bacterial proteins (defensins) by enterocytes. Epithelial cells can also produce other anti-bacterial substances, such as lysozyme or phospholipase. Probiotic bacteria have developed several adaptations and interactions with the host organism that allow them to survive and colonize the gastrointestinal tract (e.g., *Bifidobacterium longum* and *Bacteroides thetaiotaomicron* together can reduce the expression of genes responsible for fighting gram-positive bacteria. *Bifidobacterium bacilli* can also inhibit the signal stimulating the production of RegIII$\gamma$ lectin, which is a consequence of activation of TLR receptors, and *Enterococcus* has the ability to induce the expression of genes responsible for the production of IL-10, having an anti-inflammatory effect).

4.  Anti-cancer function: Bacterial enzymes play an important role in carcinogenesis. Probiotic strains can reduce the activity of carcinogens, e.g., the *Lb. acidophilus* strain causes a decrease in the activity of 1,2-dimethylhydrosine and the *Bifidobacterium longum* strain reduces the activity of 2-amino-3-methyl-limidazal (4,5-t) choline. Moreover, *Lb. casei* (LC9018) strains induce immune response mechanisms against cancer cells. In addition, the reduction of hepatic lipogenesis by probiotic strains may be useful in the treatment of cancer. Figure 2 illustrates the functions of the gut microbiota.

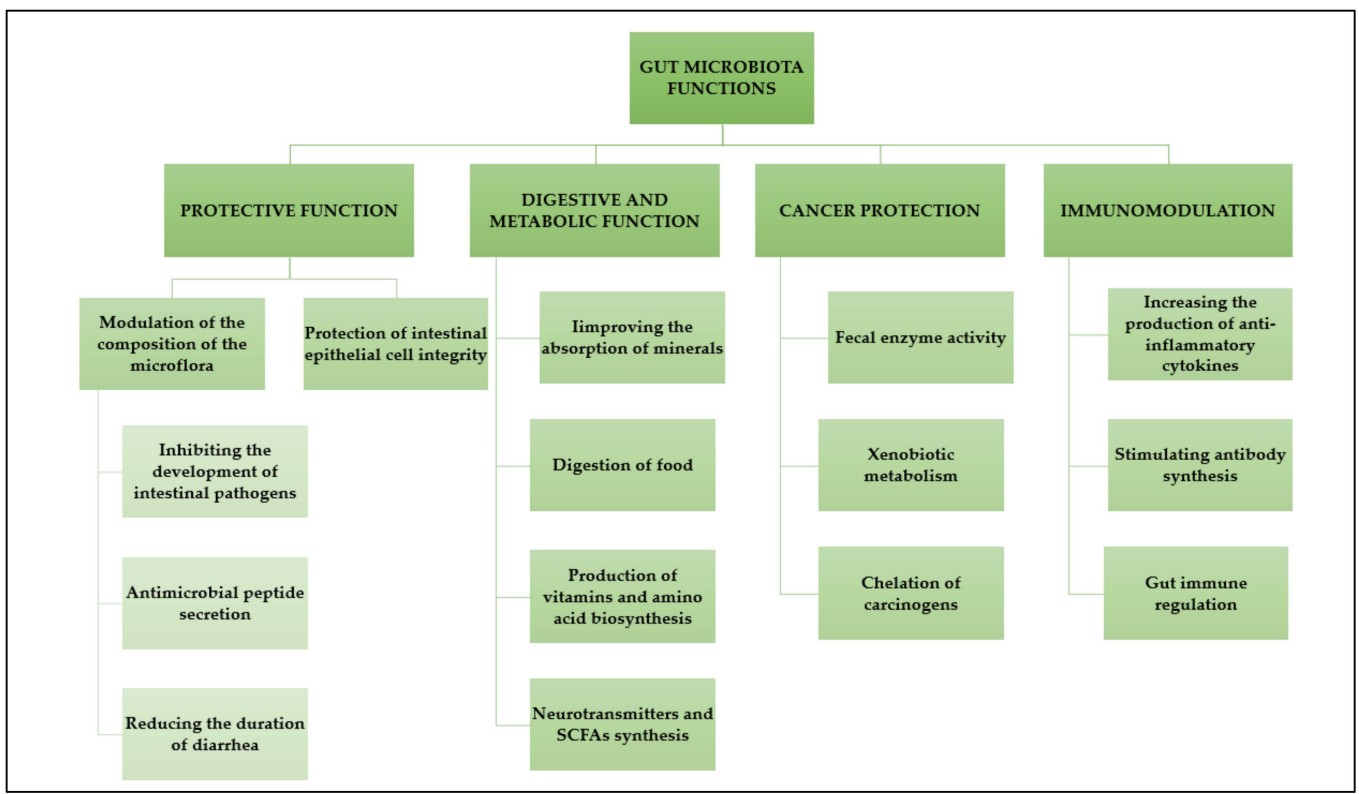

**Figure 2.** Functions of the gut microbiota [8,85–88].

### 3.3. Eubiosis and Dysbiosis

The adult intestines are in a state of eubiosis, i.e., a physiological balance regarding the amount, diversity, and composition of microorganisms inhabiting, for example, the digestive tract. The intestinal microbiota in the eubiotic state is characterized by the predominance of potentially beneficial species, mainly two types of bacteria, *Firmicutes* and *Bacteroides*, while potentially pathogenic species, such as those belonging to the type *Proteobacteria (Enterobacteriaceae)*, are present in a low percentage [41,90,91]. Dysbiosis occurs when this balance is disturbed for various reasons (e.g., antibiotic therapy or changing the diet) [41,80,81,90] and can be a consequence of growth, change in composition, or disappearance of microbiota. There are three types of dysbiosis, consisting of [92]:

1. Loss of beneficial organisms (antibiotics),
2. Excessive growth of potentially harmful organisms (infections, lack of hygiene), and
3. lLss of overall microbial biodiversity (poor diet).

The most common variation factors in the composition of the microbiota are [68,93,94] the following:

- food, food additives, and alcohol consumption—unhealthy eating habits negatively affect the composition of the gut microflora and can act as a disease-causing factor impacting metabolic pathways. A high-fat diet and meat are associated with an increased risk of Crohn's disease (CD) and ulcerative colitis (UC). The risk of inflammatory bowel syndrome (IBS) can be reduced by modulating the structure of the gut microflora and/or its metabolome with a vegetarian diet [95–98];
- antibiotics and medication—the main consequence of antibiotic treatment is the elimination of sensitive microorganisms (symbiotic bacteria) and the selection and multiplication of dysbiotic bacteria or fungi—primarily pathogenic. This imbalance of the ecosystem can lead to diarrhea due to the pathological proliferation of opportunistic endogenous pathogens, such as *Clostridium difficile* and vancomycin-resistant enterococci. Moreover, patients treated with antibiotics are more susceptible to infec-

tions caused by hexogen pathogens due to the loss of microbiota integrity and barrier function [99],

- age (in people over 70, the number of *Bacteroides* and *Bifidobacterium* decreases), gender (the effect of sex hormones), stress (under stress, the bacteria such as *Lactobacillus Bacteroides* spp. and *Clostridium* spp. decrease), lifestyle (smoking habits and drug consumption can together contribute to gut dysbiosis),
- gastrointestinal disorders and infections.

Qualitative and quantitative disturbances in the gut microbiota may lead to the development of intestinal (e.g., gastritis, diarrhea due to *Clostridium difficile*, small intestinal bacterial overgrowth (SIBO), colorectal cancer, or stomach cancer) or systemic diseases (type 1 and 2 diabetes mellitus, obesity, neurologic and psychiatric diseases, cardiovascular diseases, autoimmune diseases), underlying such chronic diseases as inflammatory bowel disease, which includes CD or ulcerative colitis [100–109]. However, a specific link between IBS and changes in the intestinal microbiome is not easy to establish, as these changes may be as much a cause as a result of the onset of disease symptoms. Studies conducted using animals that do not produce T and B lymphocytes confirmed the influence of bacteria such as *Helicobacter hepaticus* on the formation of IBS symptoms [110,111]. A change in the proportion of Gram-positive (especially *Clostridium leptum*) and Gram-negative gastrointestinal flora, particularly of the Bacteroidetes type, was also noted. An excessive amount of Gram-negative bacteria, which, due to the presence of a lipopolysaccharide wall, has a stronger effect on the immune system, may cause IBS [112–114]. Therefore, in many cases, they can be considered the disease's primary etiological factor.

Moreover, dysbiosis can also be associated with the formation of neoplastic changes. The process of carcinogenesis related to dysbiosis may be affected by several mechanisms, such as the imbalance of signals stimulating and inhibiting the development of inflammation, the cytotoxic effect and the associated excessive proliferation of epithelial cells, and the production by microorganisms of toxic, intermediate products of metabolism that can damage the cellular epithelium [59,93,100].

## 4. The Influence of Fermented Milk Products on the Microbiota Composition

Over the last decade, the human gut microbiota has gained more attention due to its beneficial effects on human health, as improving the composition of the gut microflora can prevent and support the treatment of numerous diseases. In particular, gastrointestinal disorders, such as ulcerative colitis, irritable bowel syndrome, or diarrhea, are associated with altered patterns of gut microflora [100,112,115–117].

Some studies on healthy people and people with gastrointestinal disorders suggest that fermented milk products benefit gut microflora. Most studies of fermented milk products consumed by healthy individuals have been safe, with no adverse side effects, and have beneficial effects on the gut microbiota. The influence of fermented milk products on the microbiota composition in healthy people is presented in Table 3.

Table 3. The influence of fermented milk products on the microbiota composition in healthy people.

| Fermented Milk Products Used | Type of Bacteria | Dose | Time of Intervention | Study Population | Effect | References |
|---|---|---|---|---|---|---|
| Yogurt vs. Milk fermented with yogurt cultures and *Lb. casei* vs. Nonfermented gelled milk | Both fermented products contained at least $1 \times 10^6$ CFU/g *Lb. bulgaricus*, $1 \times 10^9$ CFU/g *Sc. thermophilus*, and $1 \times 10^8$ CFU/g *Lb. casei* | 125 g/d of one of the three products | 1 week baseline period, 1 month supplementation period, and 1 week follow-up Period | Infants: 39 healthy infants (randomly assigned to one of three groups) aged 10–18 months | In the yogurt group, the number of *Enterococci* in the feces increased, and the activity of β-glucuronidase significantly decreased. The percentage of branched-chain and long-chain fatty acids significantly decreased. | Guerin-Danan et al., 1998 [118] |
| Yogurt (three different yogurts) | *Lactobacilli* $6 \times 10^7$–$2.4 \times 10^8$/g yogurt - Yogurt 1–*Lb. casei* $2.4 \times 10^8$ CFU/g yogurt, - Yogurt 2–*Lb. acidophilus* $6.0 \times 10^7$ CFU/mL yogurt and *Lb. delbrueckii* $4.5 \times 10^7$ CFU/mL yogurt, - Yogurt 3–*Lb. delbrueckii* $1 \times 10^8$ CFU/g yogurt | One serving per day depending on the study group: Yogurt 1–110 g/day, Yogurt 2–180 mL/day Yogurt 3–90 g/day | 20 days | Adults: 15 healthy adults (9 males and 6 females) were assigned to one of three groups; aged 24–46 years | The consumption of yogurts with probiotic strains was no more effective than yogurt which does not contain probiotic strains on the human fecal microbial composition. *Bacteroides* and *Prevotella* population levels and the *Clostridium coccoides Eubacterium rectale* group in fecal samples tended to change in response to ingestion, however, the change was not related to the yogurt type. The bacterial community in human feces may be altered by yogurt consumption but not related to probiotic lactic acid bacteria. | Uyeno et al., 2008 [119] |
| Strawberry yogurt with *Bifidobacterium animalis* subsp. *lactis* BB-12 vs. Yogurt without BB-12 (control group) | *Bifidobacterium animalis* subsp. *lactis* ($1 \times 10^{10}$ colony/100 mL) and *Sc. thermophilus* and *Lb. delbrueckii* subsp. *Bulgaricus* | Four fluid ounces (112 g) per day | 90 days | Children: 172 children from Washington (randomly assigned to one of two groups); aged 2–4 years | Yogurt was well tolerated in children but did not decrease absences due to illnesses in daycare. | Merenstein et al., 2011 [87] |

<center>**Table 3.** *Cont.*</center>

| Fermented Milk Products Used | Type of Bacteria | Dose | Time of Intervention | Study Population | Effect | References |
|---|---|---|---|---|---|---|
| Yogurt with *Bifidobacterium longum* BB536 vs. Ultra-high-temperature pasteurized milk | *Bifidobacterium longum* BB536 $4.27 \pm 1.25 \times 10^8$ CFU of living BB536 (more than $1.12 \pm 0.62 \times 10^8$ CFU of BB536 at the end of the study) and $1 \times 10^9$ CFU of lactic acid bacteria | One portion per day <br> - 160 g of yogurt with *B. longum* BB536 <br> or <br> - 200 mL of ultra-high temperature pasteurized milk | 8 weeks | Adults: 32 healthy adults (11 male and 21 female) from Eastern Japan (randomly assigned to one of two groups); the mean age in the yogurt group was $41.1 \pm 10.2$ years, and in the milk group, $38.6 \pm 7.5$ years | The consumption of yogurt significantly decreases enterotoxigenic *Bacteroides fragilis* in the gut microbiota. | Odamaki et al., 2012 [120] |
| Yogurt with *Bifidobacterium animalis* subsp. *lactis* BB-12 vs. Yogurt without BB-12 (control group) | *Bifidobacterium animalis* subsp. *lactis* ($1 \times 10^{10}$ CFU/100 mL) | Four fluid ounces (112 g) per day | 10 days | Adults: 40 healthy adults (16 male and 24 female) randomly assigned to one of two groups; yogurts with BB-12 ($n = 19$) and control group ($n = 21$); mean age in the yogurt group of 33 years and the control group of 29 years | *Bifidobacterium lactis* fecal levels were modestly higher in the yogurt with BB-12 group. In a small subset of participants, consuming yogurt with BB-12 activated an array of immune genes associated with regulating and activating immune cells. | Merenstein et al., 2015 [121] |
| Yogurt with *Bifidobacterium animalis* subsp. *lactis* BB-12 | *Bifidobacterium animalis* subsp. *lactis* BB-12 | Twice a day, 125 mL of yogurt in the morning and evening | 30 days | Adults: 150 healthy volunteers from Russia (no exact information about the age of the patients) | Gut microbe content showed an increase in the presence of potentially beneficial bacteria, especially the genus *Bifidobacterium*, *Adlercreutzia equolifaciens* and *Slackia isoflavoniconvertens*. Increased ability to metabolize lactose and synthesize amino acids while reducing the potential for lipopolysaccharide synthesis. | Volokh et al., 2019 [122] |

**Table 3.** *Cont.*

| Fermented Milk Products Used | Type of Bacteria | Dose | Time of Intervention | Study Population | Effect | References |
|---|---|---|---|---|---|---|
| Fermented milk product vs. Control group (without any intervention) | *Lactobacillus casei* strain Shirota at the minimum concentration of $6.5 \times 10^9$ CFU | Commercially available fermented milk product (65 mL) taken during breakfast | 6 weeks | Children: 18 healthy children; study group ($n = 6$) and control group ($n = 12$); aged 12–18 years | Fermented milk product ingestion by healthy children does not result in a more diverse and stable gut microbiota composition. | El Manouni El Hassani et al., 2019 [123] |
| Fermented milk products | *Lactocaseibacillus paracasei* strain Shirota 0.9–40 billion CFU per bottle | Intake $\geq$ 3 days/week | 1 year | Adults: 218 Japanese participants; aged 66–91 years | Stabilisation of the gut microbiota in the elderly. | Amamoto et al., 2021 [124] |
| Strawberry yogurt (control group) vs. Strawberry yogurt with strain BB-12 added pre-fermentation vs. Strawberry yogurt with BB-12 added post-fermentation vs. Capsule containing BB-12 | *Bifidobacterium animalis* subsp. *lactis* BB-12 ($\log_{10} 10 \pm 0.5 \times 10^9$ or $3.16 \times 10^9$ and $3.16 \times 10^{10}$ CFU of BB-12/ portion, in capsules $\log_{10} 10 \pm 0.5$ CFU of BB-12/capsule | 240 g yogurt/day. | 4 treatments each lasting 4 weeks, and a 2 week wash-out compliance break between treatments | Adults: 36 healthy adults; 29 finished at least one treatment period (18 females and 11 males); mean age of $28.1 \pm 0.6$ years | Consumption of yogurt with BB-12 or capsule did not significantly alter the gut microbiota composition, gut transit times, and fecal excretion of short-chain fatty acids. A significant gender effect was observed when comparing the gut microbiota. Daily consumption of BB-12 in yogurt (with strain BB-12 added pre-fermentation and post-fermentation) resulted in a higher relative abundance of *B. animalis.* | Ba et al., 2021 [125] |
| Yogurt vs. Control group (without any intervention) | Lactic acid bacteria $1.4 \times 10^9$ CFU g$^{-1}$ | 175 g of plain organic milk yogurt | 8 weeks | Adults: 52 postmenopausal women from Lativa; control ($n = 26$) and experimental group ($n = 26$); aged 44–69 years | No significant changes in the gut microbiome were related to the consumption of yogurt. Consumption of food products like grains, grain-based products, milk and milk products, and beverages (tea, coffee) is associated with differences in the composition of the gut microbiome. | Aumeistere et al., 2022 [126] |

In a study by Guerin-Danan et al., increased enterococci in infant feces were observed during 1-month supplementation of milk fermented with yogurt cultures (*Lb. bulgaricus* and *Sc. thermophilus*) and *Lb. casei* [118]. This may have been due to the survival of *S. thermophilus* during its passage through the gastrointestinal tract or to an increase in the number of endogenous enterococci. Uyeno et al. observed that the consumption by adults of three different yogurts changed the bacteroides and *Prevotella* population levels and the *Clostridium coccoides-Eubacterium rectale* group in fecal samples [119]. However, the changes did not appear to be related to the types of milk products. Additionally, yogurt consumption may alter these changes, but not the types of lactic acid bacteria. The changes in the gut microflora in adults also occurred after 8 weeks of consumption of yogurt with *Bifidobacterium longum* BB536, compared to UHT milk groups (decreases *Bacteroides fragilis*) [120]. Ten-day supplementation of yogurt with *Bifidobacterium animalis* subsp. *lactis* BB-12 increased fecal levels of *Bifidobacterium lactis* [121], and 30 days of supplementation increased *Bifidobacterium, Adlercreutzia equolifaciens*, and *Slackia isoflavoniconvertens* [122]. Moreover, Amamoto et al. observed intakes of $\geq$ three days per week of fermented milk products (*Lb. paracasei* strain Shirota) stabilized the gut microbiota in the elderly [124]. In contrast, no effect on gut microbiota was observed by El Manouni El Hassani et al. [123] after 6 weeks of supplementation of fermented milk product with *Lb. casei* strain Shirota in children, or by Aumeistere et al. after 8 weeks of supplementation of yogurt with lactic acid bacteria in adults [126].

In addition, studies of healthy adults on the influence of fermented milk products on the composition of the gut microflora have shown that they may activate an array of immune genes associated with regulating and activating immune cells. Volokh et al. reported that 30 days of supplementation of yogurt with *Bifidobacterium animalis* subp. *lactis* BB-12 increased the ability to metabolize lactose and synthesize amino acids while reducing the potential for lipopolysaccharide synthesis [122]. Levels of the transcription factor GATA3, CD80, an early inducer of T-cell proliferation, CXCL10, and pro-inflammatory TNF-$\alpha$ were upregulated at least five-fold in blood cells isolated from adults consuming yogurt with *Bifidobacterium* BB-12, compared to the control group.

The influence of fermented milk products on the gut microbiota depends on the type of fermented product and the bacterial strain used, the number of bacteria, the time of supplementation, and the study group. In addition, age and ethnicity seem to play roles, but due to the small number of studies and subjects, this topic requires further research.

The influence of fermented milk products on the gut microbiota composition in humans with gastroenterological diseases, such as diarrhea, ulcerative colitis, and irritable bowel syndrome, are presented in Table 4.

**Table 4.** The effect of fermented milk products on the microbiota composition in humans with gastrointestinal disorders.

| Fermented Milk Products Used | Type of Bacteria | Dose and Time of Intervention | Time of Intervention | Study Population | Effect | References |
|---|---|---|---|---|---|---|
| | | | Diarrhea | | | |
| Yogurt | *Sc. thermophilus* and *Lb. bulgaricus* | Individual dosage (depending on lactose) per kilogram of body weight | 4 days | Children: 9 Algerian boys with diarrhea of >1 month in duration, clinically mild malnutrition, villus atrophy, and lactose maldigestion; aged 7–29 months | Replacing milk (infant formula) with yogurt reduced lactose malabsorption and tended to improve lactose intolerance and diarrhea. | Dewit et al., 1987 [127] |
| Yogurt prepared from milk formulae vs. Milk formula | *Sc. thermophilus* and *Lb. bulgaricus* | Individual dosage per kilogram of body weight 150–180 kcal/kg/day for all foods (children aged 3–6 months received 4 servings of milk or yogurt, children aged 6–16 months received 3 servings, and children aged 12–36 months received 2 servings). | 5 days | Children: 52 children with persistent diarrhea (duration > 13 days but <29 days); randomly assigned to one of two groups; yogurt ($n = 25$) and milk ($n = 27$); age 3–36 months | Clinical failure was observed in 42% of children receiving milk and 14% receiving yogurt. Children consuming yogurt gained weight despite lower energy intake, had less liquid stools, and required less oral rehydration solution than children receiving milk. | Boudraa et al., 1990 [128] |
| Yogurt prepared from milk formulae vs. full-strength milk formulae | *Sc. thermophilus* and *Lb. bulgaricus* | 120 mL/kg body weight in seven divided feedings | 72 h | Children: 96 malnourished boys; randomly assigned to one of two groups; yogurt ($n = 47$) and milk ($n = 49$); age 4–47 months | The treatment failure rate was similar in both groups. Children who consumed milk had more weight gain at the end of the study and after recovery. Yogurt for malnourished children with acute diarrhea has no significant clinical benefit over milk. | Bhatnagar et al., 1998 [129] |

**Table 4.** *Cont.*

| Fermented Milk Products Used | Type of Bacteria | Dose and Time of Intervention | Time of Intervention | Study Population | Effect | References |
|---|---|---|---|---|---|---|
| Standard yogurt vs. Fermented milk with yogurt cultures and *Lb. casei* vs. Jellied milk (control group) | *L casei* $1 \times 10^8$ CFU/mL | One of three products 125 g or 250 g according to age | Three periods of 1 month, followed by 1 month without intervention | Children: 287 children with acute diarrhea over a 6-month observation period; mean age of $18.9 \pm 6$ months | The incidence of diarrhea was not different between the groups. The severity of diarrhea significantly decreased with the supplementation of *L. casei* fermented milk compared with the jellied milk. | Pedone et al., 1999 [130] |
| Pasteurized yogurt and routine hospital care vs. Routine hospital care (control group) | *Lb. bulgaris* $5 \times 10^4$ /mL and *Sc. thermophilus* $5 \times 10^4$/mL | 15 mL/kg/day | Until hospital discharge | Children: 80 children with moderate dehydration and acute non-bloody, non-mucoid diarrhea; randomly assigned to one of two groups; yogurt (*n* = 40) and control group (*n* = 40); aged 6–24 months | Children receiving yogurt observed a reduction in the frequency of diarrhea, fewer days in the hospital, and more weight gain compared to the control group. | Pashapour and Iou, 2006 [131] |
| Fluid yogurt prepared from commercial yogurt vs. Lyophilized *Saccharomyces boulardii* | $1 \times 10^7$ CFU/100 mL of *Lb. bulgaricus* and *S. thermophilus* (yogurt group) | Yogurt group: 15 mL twice a day for children < 2 years and 30 mL twice a day for children ≥ 2 years Lyophilized *Saccharomyces boulardii* group: 250 mg twice a day in children ≥ 2 years and 125 mg twice a day in children < 2 years of age | Until the resolution of the diarrhea | Children: 55 children with diarrhea; randomly assigned to one of two groups; yogurt (*n* = 27) and lyophilized *Saccharomyces boulardii* (*n* = 28); age 5 months–16 years | The effect of yogurt was comparable with that of lyophilized *Saccharomyces boulardii* in the treatment of acute diarrhea | Eren et al., 2010 [132] |

**Table 4.** *Cont.*

| Fermented Milk Products Used | Type of Bacteria | Dose and Time of Intervention | Time of Intervention | Study Population | Effect | References |
|---|---|---|---|---|---|---|
| | | | Ulcerative Colitis (UC) | | | |
| Bifidobacteria-fermented milk vs. control group | $1 \times 10^{10}$ CFU of *Bifidobacterium breve*, and *Bifidobacterium bifidum*, and *Lb. acidophillus* YIT 0168 | 100 mL/day | 1 year | Adults: 21 patients with UC remission; randomly assigned to one of two groups; study group (*n* = 11), control group (*n* = 10); age 39–60 years | Significant reduction in exacerbation of symptoms after bifidobacteria fermented milk supplementation. Reduction in the percentage of *Bacteroides vulgatus* and *luminal butyrate* and good recovery of probiotic strains in the stools. Increases in protein and albumin levels. | Ishikawa et al., 2003 [133] |
| Bifidobacteria fermented milk vs. Fermented milk without live bifidobacteria (control group) | $\geq 1 \times 10^{10}$ CFU of *Bifidobacterium bifidum* strain Yakult and *Lb. acidophillus* strain | 100 mL/day | 12 weeks | Adults: 20 patients with active UC, randomly assigned to one of two groups; study group (*n* = 10), control group (*n* = 10); mean age of 30.2 years for the study group and 33.7 years for the control group | Increase in probiotic strains and butyrate in the feces. Improved clinical activity index; endoscopic activity index and histological scores compared to the control group. | Kato et al., 2004 [134] |

**Table 4.** *Cont.*

| Fermented Milk Products Used | Type of Bacteria | Dose and Time of Intervention | Time of Intervention | Study Population | Effect | References |
|---|---|---|---|---|---|---|
| Fermented milk product (Cultura) | $1 \times 10^8$ CFU/mL milk *Lb. acidophilus* La-5 and *B. lactis* BB-12 | 500 mL | 4 weeks | Adults: three groups: UC group with ileal-pouch-anal-anastromosis (*n* = 51, mean of age 40 years), familial adenomatus polyposis with ileal-pouch-anal-anastromosis (*n* = 10, mean of age 35 years) and UC with ileorectal anastromosis (*n* = 6, mean of age 42 years) | Increased number of *lactobacillus* and *bifidobacterium* in the UC patients with ileal-pouch-anal-anastromosis and remained increased one week after intervention. No significant changes in blood tests (antinuclear antibody and antineutrophil autoantibodies), fecal fungi and fecal pH. | Laake et al., 2005 [135] |
| Fermented milk products with *Bifidobacterium breve* strain Yakult | $1 \times 10^{10}$ CFU of *Bifidobacterium breve* *Lb. acidophilus* and $1 \times 10^9$ CFU of *Lb. acidophilus* | One pack (100 mL) of commercial *B. breve* strain Yakult fermented milk (Mil–Mil) | 48 weeks | Adults: 195 Japanese patients with quiescent UC; study group (*n* = 98) and placebo group (*n* = 97); aged 20–70 years | *Bifidobacterium breve* strain Yakult did not affect the time to relapse in UC patients compared with the placebo group. | Matsuoka et al., 2018 [136] |
| | | | Irritable bowel syndrome (IBS) | | | |
| Probiotic fermented yogurt drink vs. Placebo (the same product without lactic acid fermented bacteria) | $4 \times 10^9$ CFU of *Lb.* sp. HY7801, *Lb. brevis* HY7401, and *Bifidobacterium longum* HY8004 | One bottle (150 mL) of a probiotic yogurt drink, 3 times/day, within 10 min after breakfast, lunch, and dinner | 8 weeks | Adults: 74 IBS patients from the Republic of Corea; randomly assigned to one of two groups; study group (*n* = 37) and placebo group (*n* = 37); range age of 33 years | The amount of *Lactobacilli* species, which were included in the yogurt drink, significantly increased in the feces of IBS patients receiving treatment. Serum glucose and tyrosine levels in IBS patients were normalized to those of healthy individuals in the study group. | Hong et al., 2011 [137] |

Studies on the effect of supplementation with fermented milk products in children with diarrhea have shown that they do not significantly affect treatment, as replacing milk formulae with yogurts prepared based on milk formulae did not significantly affect the course of diarrhea compared to children receiving milk formulae. However, some studies have shown that the administration of yogurt with *Lb. casei* [130] or *Lb. Bulgaris* [131] may reduce the frequency of diarrhea. It is worth noting that the studies mainly concerned infants and children up to 3 years of age. There are currently no studies evaluating the effect of fermented dairy consumption on the gut microbiota in adults with diarrhea. Most studies lacked a detailed description of the patient's diet, so subsequent studies should collect a detailed nutritional history, which would allow for the possible determination of failures in using fermented milk products. It is also worth considering research on an older age group, where the gut microbiota is more stabilized.

The effect of fermented milk products on UC patients was varied, with some studies confirming the effect of fermented milk products on reducing exacerbation symptoms, and some showing no effect on the course of UC. Ishikawa et al. observed a reduced number of *Bacteroides vulgatus* and *luminal butyrate* in fecal samples, as well as increased serum protein and albumin levels [133]. Kato et al. observed numbers of *Bifidobacterium breve* and *B pseudocatenulatum* while improving clinical parameters [134]. However, Laake et al. found no changes in serum levels of antinuclear antibodies and anti-neutrophil autoantibodies despite increasing numbers of *Lactobacillus* and *Bifidobacterium* in fecal samples [135]. Matsuoka et al. reported that 48 weeks of supplementation of fermented milk products with *Bifidobacterium breve* and *Lb. acidophilus* did not affect relapse in UC patients, compared to the control group [136]. These differences may relate to the use of different bacterial strains, study duration, or the study groups (stage of disease, age, ethnicity), and to the parameters used for assessment.

The IBS effect of fermented milk products is not well known. Based on the available literature review, one randomized study determined the effect of consuming fermented milk products on patients with IBS. Hong et al. showed that 8 weeks of supplementation of yogurt drinks with *Lb*. sp. HY7801, *Lb. brevis* HY7401, and *Bifidobacterium longum* increased *Lactobacilli* species in fecal samples, compared to the control group [137]. Interestingly, this study suggests that elevated serum glucose and tyrosine levels in IBS patients may be reversed by probiotic supplementation and play a role in the pathogenesis of this disease.

## 5. Conclusions

The consumption of fermented milk products may have a beneficial effect on the microbial biodiversity of the gut microbiota, stabilize the gut microbiota in the elderly, and support the treatment of dysbiosis. Based on current research, there is insufficient evidence of fermented milk products' beneficial effects on treating gastrointestinal diseases, such as ulcerative colitis and irritable bowel syndrome. Although the results of studies evaluating the effect of fermented milk products on the occurrence of diarrhea are very promising (reduction in the frequency of diarrhea, fewer days in the hospital), the topic still requires further research. Modulating gut microbiota with fermented milk products requires further study to optimize the microorganism used, the dose, and the duration.

Fermented milk products rich in probiotics and postbiotics exert a beneficial effect on gut microbiota and may be developed into functional food products with immune modulating effects.

**Author Contributions:** Conceptualization, A.O., M.D. and S.D.-C.; data collection and interpretation, A.O., M.D., P.K., K.D., J.P. and S.D.-C.; writing—original draft preparation, A.O., M.D. and S.D.-C.; writing—review and editing, K.D., J.P. and S.D.-C.; supervision, S.D.-C.; project administration, S.D.-C. All authors have read and agreed to the published version of the manuscript.

**Funding:** This research received no external funding.

**Institutional Review Board Statement:** Not applicable.

**Informed Consent Statement:** Not applicable.

**Data Availability Statement:** No new data were created or analyzed in this study. Data sharing is not applicable to this article.

**Conflicts of Interest:** The authors declare no conflict of interest.

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
