# Peer review of "The Role of Fermented Dairy Products on Gut Microbiota Composition"

_fermentation, doi:10.3390/fermentation9030231_

Round 1
Reviewer 1 Report
I found the paper to be overall well written, complete and well described. However, it could be improved by means of little actions:
Lines 103 and 110: replace “Villi” with “Viili”
Line 141: delete “of the digestive content”
Line 146: delete “of the content”
Line 272: I think it is more accurate to talk about “variation factors” than “factors causing disturbances”
Lines 287 and 288: I don’t understand to what do your refer with these items (age, gender, stress, lifestyle, gastrointestinal disorders and infections) ¿are they variation factors of the microbiota composition as food, food additives, antibiotics, etc.? In this case the items must be written with the same format.
Figure 1: the size of the letters is too small to be read
Figure 1: the title will be more complete if you replace “microbiota” with “gut microbiota”
Reviewer 2 Report
Prezentowana przez źródÅ‚o publikacja przedstawia interesujÄ…cÄ… tematykÄ™ i dogÅ‚Ä™bnie przedstawiÅ‚a problematykÄ™ fermentowanych produktów mlecznych.
ZgÅ‚oszone w Znajomym uwagi dotyczÄ…ce gÅ‚ównych przyczyn redakcyjnych. Uwagi lub rekomendacje sÄ… opisane w pliku pdf.
Prezentowany artykuł jest zrozumiały i napisany poprawnie.
Praca ma wysoką wartość naukową.

Reviewer 3 Report
Manuscript entitled The role of fermented dairy products on gut microbiota composition, reviews the types of fermented dairy products and their microflora. Presented here is a summary of research on the consumption of dairy products on the gastrointestinal microflora of both healthy people and those with various ailments such as diarrhea, ulcerative colitis and irritable bowel syndrome. Below please find my review along with tips and suggestions for this manuscript:
- L 31-39 The authors make it clear that the above literature review is a systematic review and provide the methodology for its creation, inclusion and exclusion criteria (original articles, English language). In my opinion, it would still be useful to add information on how many articles were found after the initial search, how many were rejected (for various reasons) or what % of all articles originally searched were finally included in the development of this manuscript.
- L 97 The title of Table 1 should be changed, as it indicates that the composition of cow's, goat's, sheep's and buffalo's milk is in Table 1, nevertheless mare's, camel's, donkey's and yak's milk are also placed in the table. I suggest changing the title to "Composition of milk from various animals."
- Table 1 - niacin has no unit given.
- Table 2 - With some microbial names, the species names are capitalized, please correct this.
- Section 3 and subsection 3.1 - Section 3 and subsection 3.1 should be combined into one, or subsection 3.1 should be renamed. In my opinion, the current name is not appropriate for the content presented in subsection 3.1. A more fitting title would be, for example, “Variability in gut microbiota composition” or “Factors affecting variability in gut microbiota composition”.
- L 191 -I'm not sure if the term "mode of delivery" is the correct phrase in the context of the type of childbirth (natural or by cesarean section).
- In my opinion, Table 3 is unnecessary because it repeats information previously contained in the text.
- Figure 1 is completely unreadable. I suggest separating it into several separate diagrams showing the functions of the intestinal microbiota.
- L 386 Lactobacillus should be in italics.
- The conclusion section is too short. Please summarize how and if indeed the consumption of fermented dairy products affects the intestinal microflora of a healthy person. Are there any specific groups of dairy products that change the intestinal microflora of healthy people and people with various ailments particularly strongly, compared to others? What are the conclusions drawn from this review and recommendations for the future in the context of dairy consumption?
- Please pay more attention to the names of journals in the literature list and correct according to the guide for authors. Please be consistent and use abbreviations of journal names instead of full names, e.g. lines 447, 488, 502, 505, 521, 564, etc.
Reviewer 4 Report
1. Abstract should revise, need to write about the gap, need organization and the aim of this review
2. Introduction is very poor, need to write the what is the need of this review? Still now how many reviews are present in this area? and what is the gap and need for the review?
3. What is the need to write the section 2.1 as it is the milk composition?
4. In table 1 given exact values, but better to give the range
5. Line 98, what is the mean of “Tr – trace amount”? What is the amount considered?
6. Section 2.2 need to discuss wider products like the use of other ingredients like fruits and others.
7. In Table 2, what is the character means in the last column? As the character is a term with broad meaning
8. Section 3, The microbiota is an opt term? Better to use the intestinal microbiota?
9. In “3.1. Composition of the gut microbiota”, better to give which microbe is influence by different factors like increase, decrease etc
10. Section 3.2 is vague, better to discuss the scientific mechanism in each
11. Figure 1. Must revise, the color, font size, and presentation style.
12. In lines 274 and 280 it is side headings heading? Must be in bold.
13. What is the need of table 4, it contains general information only.
14. In section 4, line 318 needs to work on the what are effects of the other components, for instance, there is a pre and probiotics effects will be there hence check this and explain them. This is very important in the case of fruit yogurts.
15. In table 6, better mention the age of the children’s in each study reported.
16. Conclusions are very poor.
General comments:
1. Editorial issues are persistent hence check these issues
2. Typographical errors are persistent
3. Check for the English language issues
4. In the introduction presented the methodology followed for this work, hence, the shift in the appropriate place.
5. Scientific quality of the write-up should improve
6. the structural organization should be revised
Round 2
Reviewer 4 Report
All the comments are handled properly
